# A FAST, RELIABLE, AND SECURE PROGRAMMING LANGUAGE FOR LLM AGENTS WITH CODE ACTIONS

## ABSTRACT

Modern large language models (LLMs) are often deployed as *agents*, calling external tools adaptively to solve tasks. Rather than directly calling tools, it can be more effective for LLMs to write code to perform the tool calls, enabling them to automatically generate complex control flow such as conditionals and loops. Such *code actions* are typically provided as Python code, since LLMs are quite proficient at it; however, Python may not be the ideal language due to limited built-in support for performance, security, and reliability. We propose a novel programming language for code actions, called QUASAR, which has several benefits: (1) automated parallelization to improve performance, (2) uncertainty quantification to improve reliability and mitigate hallucinations, and (3) security features enabling the user to validate actions. LLMs can write code in a subset of Python, which is automatically transpiled to QUASAR. We evaluate our approach on the ViperGPT and CaMeL agents, applied to the GQA visual question answering and AgentDojo AI assistant datasets, demonstrating that LLMs with QUASAR actions instead of Python actions retain strong performance, while reducing execution time by up to $56\%$, improving security by reducing user approvals by up to $53\%$, and improving reliability by applying conformal prediction to achieve a desired target coverage level.

## 1 INTRODUCTION

Large language models (LLMs) have recently demonstrated remarkable general reasoning capabilities. To leverage these capabilities to solve practical tasks, there has been significant interest in *LLM agents*, where the LLM is given access to *tools* that can be used to interact with an external system. The LLM can autonomously choose when to use different tools to help complete a task given by the user. Tools include functions to read and edit files (Yang et al., 2024), access to knowledge sources such as databases (Lewis et al., 2021), external memory to store information across different interactions (Liu et al., 2024b; Maharana et al., 2024), and access to user input/output devices such as mouse, keyboard, and screen (Anthropic, 2025).

An effective strategy in practice is to provide these tools in the form of software APIs, and then let the LLM write code that invokes these APIs (Wang et al., 2024b; Surís et al., 2023; Trivedi et al., 2024; Debenedetti et al., 2025); we refer to these systems as LLM agents with *code actions*. This strategy enables the LLM to write code that includes control flow to facilitate more complex interactions, such as automating iterative tasks by writing loops. For example, ViperGPT gives the LLM access to external tools such as object detectors to perform image question answering (Surís et al., 2023), and AppWorld gives the LLM access to a rich variety of smartphone app APIs to enable it to help the user automatically configure their device (in a simulation) (Trivedi et al., 2024).

A natural question is what the ideal programming language is for code actions. Python has become the standard choice due to the presence of a large existing ecosystem of software libraries; furthermore, due to the large amount of Python code in most LLM pretraining corpora, LLMs have been shown to be proficient at writing Python code (Zhuo et al., 2025; Puri et al., 2021; Shypula et al., 2024; 2025).

However, there are also a number of drawbacks of using Python. It is a highly dynamic language, making it difficult to provide assurances that the generated code is safe to execute. It is also challenging to optimize, when many agent workflows exhibit significant potential for parallelism; for instance, programs generated by ViperGPT often call multiple APIs that could be executed in parallel. In addition, agents may call other models, which are themselves prone to hallucination.

While conformal prediction (Vovk et al., 2005) can mitigate this for an individual model call by returning a set, the rest of the agent's program must then be executed with a set of values rather than a single concrete value. Python cannot do this kind of set-based execution. As a consequence, there is a unique opportunity to rethink the programming language that forms the basis of code actions.

We propose a novel agent language, QUASAR (for QUick And Secure And Reliable) that combines several promising recent ideas from the programming languages literature. The key idea is to separate *internal computation* from *external actions*. Specifically, QUASAR has a pure, functional "core language" based on lambda calculus, with side effects isolated in "external calls". Internal computations are things like executing the "then" branch of an "if" statement when the condition is true. External actions are things like executing shell programs or making requests to remote APIs. This separation provides several benefits: (1) it enables QUASAR to make use of recently proposed techniques for automatically executing external calls in parallel when possible (Mell et al., 2025), (2) it can enforce whitelists on external calls to ensure that undesirable APIs are not executed without user permission, and it can efficiently ask the user for approval in batches, and (3) it can incorporate recent techniques for uncertainty quantification in neurosymbolic programs (Ramalingam et al., 2024).

A key challenge is that unlike Python, LLMs have never seen QUASAR code and therefore do not know how to write code in this language. Rather than directly teach them QUASAR, we propose an alternative strategy where we first implement a transpiler from a subset of Python to QUASAR, and then have the LLM generate Python code in this subset. Then, whenever the LLM writes code to be executed, we translate it to QUASAR and execute it using the QUASAR interpreter instead.

**Contributions.** (1) We introduce QUASAR, a novel programming language for LLM agent actions. (2) We propose a generation strategy for QUASAR code by first asking the LLM to generate code in a subset of Python, and then transpiling that to QUASAR. (3) We experimentally demonstrate that our generation strategy achieves task performance comparable to standard Python generation. (4) We experimentally demonstrate the utility of QUASAR, reducing execution time when possible by 56%, improving security by reducing user approvals when possible by 53%, and improving reliability by applying conformal prediction to achieve a target error rate.

## 2 RELATED WORK

With the promising capabilities of LLMs, numerous studies have explored their use as autonomous agents (Wang et al., 2024a; Huang et al., 2024; Yang et al., 2024). Early efforts, such as Chain-of-Thought prompting (Jason Wei, 2023), demonstrated that providing in-context reasoning examples can significantly enhance LLM reasoning abilities. Recognizing the tendency of LLMs to produce hallucinations, subsequent work like Retrieval-Augmented Generation (RAG) (Lewis et al., 2021) and Dense Passage Retrieval (DPR) (Karpukhin et al., 2020) introduced mechanisms to incorporate external knowledge bases, using retrieved information to improve model accuracy and reliability.

Building on this idea, ReAct (Yao et al., 2023b) extends the role of external resources by providing LLMs with access to executable APIs and external tools, enabling them to perform simple tasks through API calls. While these approaches primarily guide agentic behavior via natural language, recent works such as CodeAct (Wang et al., 2024b), ViperGPT (Surís et al., 2023), AppWorld (Trivedi et al., 2024), and CaMeL (Debenedetti et al., 2025) take a step further by instructing LLMs to generate executable Python code as agent actions. This transition from natural language to code-based actions has demonstrated improved task performance and greater flexibility.

However, despite these advancements, several challenges remain for LLM agents. These include security and privacy risks (He et al., 2024; Andriushchenko et al., 2025), persistent hallucination issues (Liu et al., 2024a; Li et al., 2024a), and concerns over computational efficiency (Yao et al., 2023a). Recent work has proposed addressing the security vulnerabilities by analyzing dataflows in agent-generated code, focusing on a restricted subset of Python (Debenedetti et al., 2025). However, they do not offer performance or reliability improvements, and their approach does not support asking for batch user approval. Other work has addressed conformal prediction of functional programs with neural components (Ramalingam et al., 2024) and automatic parallelization of functional programs (Mell et al., 2025). Though we draw on insights from this work, neither considers programs generated by LLM agents or the imperative features of languages like Python.

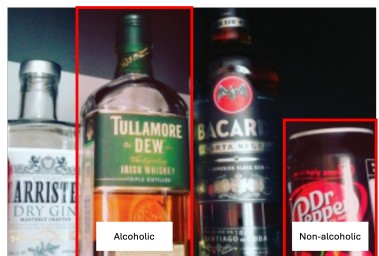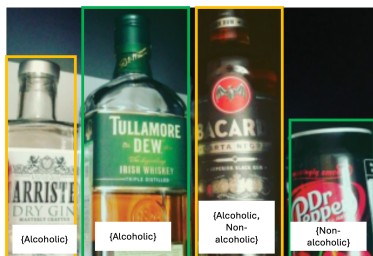

**Question:** Is there an alcoholic drink in this image?

Figure 1: Illustrative example of an image and a natural language question about that image. We show predictions of both the original object detector (left) and the conformal detector (right). For the latter, the green boxes are identifed as being definitely in the image whereas the yellow boxes may or may not be in the image. The program $P_1$ in Figure 2 answers this question for its input image.

## 3 QUASAR PROGRAMMING LANGUAGE

We first describe the syntax and semantics for QUASAR programs; then, we provide details on how QUASAR improves security, performance, and reliability (summarized in Algorithm 1). We show a running example in Figure 2 for the problem in Figure 1.

### 3.1 SYNTAX AND SEMANTICS

A QUASAR program $P \in \mathcal{P}$ consists of standard syntatic constructs such as conditionals, loops, and function calls. The execution of a QUASAR program $P \in \mathcal{P}$ is expressed as a set of *rewrite rules* $\mathcal{R}$. If a rule $R \in \mathcal{R}$ is applicable to $P$, then it transforms $P$ into a new program $P'$, which we denote by $P \xrightarrow{R} P'$. In general, there may be multiple possible programs $P'$ satisfying $P \xrightarrow{R} P'$, for instance, if the rule $R$ is applicable to different parts of $P$. If there is any rewrite $R$ mapping $P$ to $P'$, then we simply write $P \rightarrow P'$. We give the full set of rewrite rules in Figure 6 in Appendix A.

There are two kinds of rewrite rules: internal rules $\mathcal{R}_{\text{int}}$ and external rules $\mathcal{R}_{\text{ext}}$. Internal rules do not have *effects*, meaning they do not have consequences external to the program, including network calls, system calls, calls to external APIs, or even printing. Internal rules perform transformations such as substituting variables, unrolling loops, and resolving conditionals; these rules are applicable if the necessary values are constants (e.g., a conditional where the predicate is `True` or `False`).

For example, in program $P_2$ in Figure 2, the list in the for loop is a constant value `[patch1, patch2]`, so QUASAR applies a rule to unroll the for loop, resulting in $P_3$. Similarly, in $P_4$, it can apply rewrite rules to rewrite the predicate `"no" == "yes"` to `False` and the predicate `"yes" == "yes"` to `True`, after which it can rewrite the conditionals to obtain $P_5$.

There is only one external rule $\mathcal{R}_{\text{ext}} = \{R_{\text{ext}}\}$. This rule is designed to enable calls to *external functions* $f \in \mathcal{F}_{\text{ext}}$. Unlike a typical function, which is implemented as QUASAR code, an external function is implemented in Python; thus, external functions can perform desirable effects such as printing a value or calling an LLM to obtain its output. An *external call* in program $P$ is a statement $S = y \leftarrow f(x_1, ..., x_k)$ that calls an external function $f \in \mathcal{F}_{\text{ext}}$.

QUASAR executes external calls as soon as all their arguments are available. In more detail, an external call $S = y \leftarrow f(x_1, ..., x_k)$ in a program $P$ is *dispatchable* if all of $x_1, ..., x_k$ are values (e.g., `0`, `True`, or `"foo"`; recall that variables become values as the program is incrementally rewritten). As QUASAR performs rewrites, it keeps track of the currently executing external calls $(S, B) \in E$, where $S$ is a pointer to the external call in the current program $P$ (preserved by rewrites) and $B$ is a pointer to a value that is initially $\varnothing$ but is eventually set to the output of the external function. After a rewrite $P \rightarrow P'$, QUASAR identifies all the dispatchable external calls $S$ in $P'$ that are not yet in $E$; for each $S = y \leftarrow f(x_1, ..., x_k)$, it executes $f$ on $x_1, ..., x_k$ in a separate thread $T$, and adds the pending calls $(S, B)$ to $E$. The thread $T$ is also given $B$; once it finishes executing the external function $f$, it writes the output of $f$ to $B$ and terminates. Then, QUASAR applies rewrite rule $R_{\text{ext}}$ to the current program (which may no longer be $P'$) to substitute the value in $B$ into the program.

$$P_1 = \begin{array}{l}\texttt{drink\_patches = image\_patch.find("drink")} \\ \texttt{found = False} \\ \texttt{for drink\_patch in drink\_patches:} \\ \quad \texttt{if drink\_patch.simple\_query("Does this have alcohol?"):} \\ \qquad \texttt{found = True} \\ \texttt{return found}\end{array}$$

---

$$E_1 = \{(\texttt{image\_patch.find("drink")}, \varnothing)\} \rightsquigarrow E_2 = \{(\texttt{image\_patch.find("drink")}, \texttt{[patch1, patch2]})\}$$

---

$$P_2 = \begin{array}{l}\texttt{found = False} \\ \texttt{for drink\_patch in [patch1, patch2]:} \\ \quad \texttt{if drink\_patch.simple\_query("Does this have alcohol?"):} \\ \qquad \texttt{found = True} \\ \texttt{return found}\end{array}$$

---

$$P_3 = \begin{array}{l}\texttt{found = False} \\ \texttt{if patch1.simple\_query("Does this have alcohol?") == "yes":} \\ \quad \texttt{found = True} \\ \texttt{if patch2.simple\_query("Does this have alcohol?") == "yes":} \\ \quad \texttt{found = True} \\ \texttt{return found}\end{array}$$

---

$$E_3 = \{(\texttt{patch1.simple\_query(...)}, \varnothing), (\texttt{patch2.simple\_query(...)}, \varnothing)\} \quad \rightsquigarrow$$
$$E_4 = \{(\texttt{patch1.simple\_query(...)}, \texttt{"no"}), (\texttt{patch2.simple\_query(...)}, \texttt{"yes"})\}$$

---

$$P_4 = \begin{array}{l}\texttt{found = False} \\ \texttt{if "no" == "yes"} \\ \quad \texttt{found = True} \\ \texttt{if "yes" == "yes"} \\ \quad \texttt{found = True} \\ \texttt{return found}\end{array}$$

---

$$P_5 = \texttt{return True}$$

Figure 2: Given program $P_1$ for the question in Figure 1, QUASAR may execute it as follows. First, is immediately dispatches `image_patch.find("drink")`, resulting in execution set $E_1$. This external call finishes running and returns `[patch1, patch2]`, resulting in execution set $E_2$, after which QUASAR applies $R_{\text{ext}}$ to substitute this value into $P_1$ to obtain $P_2$. Then, QUASAR applies an internal rule to unroll the for loop in $P_2$ to obtain $P_3$. It immediately dispatches both `patch1.simple_query(...)` and `patch2.simple_query(...)` resulting in execution set $E_3$. As before, these external calls finish running and return `"no"` and `"yes"`, respectively, yielding $E_4$, so QUASAR applies $R_{\text{ext}}$ twice (once for each external call) to substitute these values into $P_3$ to obtain $P_4$. Finally, QUASAR applies additional internal rules to simplify the conditionals in $P_4$, resulting in terminal program $P_5$.

For example, in Figure 2, given the initial program $P_1$, QUASAR immediately dispatches the external call `image_patch.find("drink")` in a separate thread, leading to execution set $E_1$. When this thread finishes, it will write the result `[patch1, patch2]` to $\varnothing$, resulting in $E_2$. This allows $R_{\text{ext}}$ to be applied to $P_1$, obtaining $P_2$. Similarly, as soon as QUASAR rewrites $P_2$ to $P_3$, it dispatches two external calls `patch1.simple_query(...)` and `patch2.simple_query(...)`, resulting in $E_3$; these execute and return `"no"` and `"yes"`, respectively, resulting in $E_4$. Finally, QUASAR applies $R_{\text{ext}}$ twice to substitute these values into $P_3$, resulting in $P_4$. The general approach is given in Algorithm 1.

A program $P$ is *terminal* if no rules are applicable to $P$, and there are no pending external calls. Assuming each external call only depends on its inputs, then it can be shown that any sequence of rule applications results in the same set of external calls, and therefore the same effects. The order in

**Algorithm 1** Pseudocode for the QUASAR interpreter. At each iteration, it validates the current set of external calls with the user, and then executes them. It then rewrites $P$ as much as possible (including waiting for pending external calls to finish running), until it is stuck. Then, it repeats the process until $P$ cannot be rewritten any further, at which point it returns the result.

> **function** RUNQUASAR($P$)
>     **while** $P$ has dispatchable external calls **or** $P$ is not terminal **do**
>         Identify dispatchable external calls $\{S\}$ in $P$
>         Query user to validate $\{S\}$, and terminate execution if rejected
>         Dispatch all external calls in $P$ and add to a set $E$
>         $P \leftarrow$ RUNINTERNAL($P, E$)
>     **return** $P$
> **function** RUNINTERNAL($P, E$)
>     **while** $E \neq \varnothing$ **or** $P$ is not terminal **do**
>         **if** there exists $(y \leftarrow f(x_1, ..., x_k), B) \in E$ such that $B \neq \varnothing$ **then**
>             apply $R_{\text{ext}}$ to $P$ to substitute $B$ in for $y$
>         **else if** there exists a rule $R \in \mathcal{R}_{\text{int}}$ that is applicable to $P$ **then**
>             apply $R$ to $P$
>     **return** $P$

which the effects happen may be different depending on the sequence of rules applied; dependencies can be enforced by inserting arguments and return values into the relevant external calls, similar to how a pseudorandom number generator can be added to code for deterministic execution.

Because of this property (and assuming external calls depend only on their inputs), the QUASAR interpreter can apply rules to $P$ in any order. The specific strategy it employs is to first minimize the amount of interaction with the user required to validate the external calls it makes, while maximizing performance. These details are discussed in Sections 3.3 & 3.2, respectively.

There are two key benefits of this design of QUASAR. First, it decouples side effects (external) from the pure computation (internal). For instance, any internal rewrite rules cannot pose security issues by construction, since they do not have any effects on the world (other than consuming computational resources to run); thus, we only need to worry about external calls when considering potential security issues. Further, this separation makes it much easier to implement conformal semantics for QUASAR than it would be for Python. Second, because the rules can be applied in any order, execution can continue while waiting for time-consuming external calls to finish running. This is useful both for parallelizability and for reducing the number of user interaction required to validate external calls. We describe these benefits in more detail below.

### 3.2 PERFORMANCE VIA PARALLEL EVALUATION

The strategy QUASAR uses to minimize the number of rounds of interaction for security automatically parallelizes external calls, since all external calls in $P$ are dispatched simultaneously in the RUN-QUASAR routine. The actual ability to expose parallelism comes from the design of the QUASAR language and its internal rewrite rules. Intuitively, because QUASAR programs are interpreted using rewrite rules, a statement can be "executed" as soon as the relevant program variables are substituted with constants. This property enables QUASAR to execute statements out-of-order. For example, in program $P_3$ in Figure 2, the statement `patch2.simple_query(...)` can be evaluated even though previous statements have not yet been evaluated, since all of the arguments in this external call (the image patch `patch2` and the string `"Does this have alcohol?"`) are constants. As a consequence, this external call can be dispatched in parallel with `patch1.simple_query(...)`, which significantly improves performance compared to ordinary sequential execution in Python.

### 3.3 SECURITY VIA DYNAMIC ACCESS CONTROL

We consider a standard security model based on access control (Sandhu & Samarati, 1994; Sandhu, 1998), where the user must approve the execution of effects. Because effects are isolated in external calls, we only need to ensure that external calls are consistent with the user's desired security policy.

For instance, a smartphone user might give an app access to resources such as the user's location and the ability to send emails, in which case the app would only be allowed to access these resources.

In QUASAR, access to certain external functions can be granted ahead of time; alternatively, the user can dynamically approve each external call made by the program. A key challenge with dynamic access control is minimizing the number of rounds of interaction with the user; frequent interruptions can lead to poor usability. Thus, QUASAR is designed to "collect" as many external calls as possible and then query the user to confirm all of them. If rejected, execution terminates; otherwise, the external calls are all dispatched in parallel and execution proceeds. This algorithm is summarized in the RUNINTERNAL subroutine in Algorithm 1, which performs as many rewrites of the current program $P$ as possible (including both applying internal rules as well as handling previously-dispatched external calls). It returns once $P$ cannot be rewritten any further, in which case the main routine RUNQUASAR queries the user to validate all the external calls in $P$, and then dispatches all of these calls in parallel. This loop continues until $P$ is terminal. For example, in Figure 2, QUASAR asks the user for permission to make the external call `image_patch.find("drink")` in $P_1$, but then is able to batch the permission requests for `patch1.simple_query(...)` and `path2.simple_query(...)` in $P_3$.

## 3.4 RELIABILITY VIA CONFORMAL SEMANTICS

We also implement *conformal semantics* in QUASAR for uncertainty quantification. Conformal prediction is a popular technique for quantifying the uncertainty of individual blackbox machine learning models by modifying a given model to output a set of labels instead of a single label. For example, an image classification model might output a set of plausible class labels instead of just the most likely one. When QUASAR makes external calls to other machine learning models, we may want to quantify the uncertainty of these models, and then keep track of how this uncertainty propagates through the program. Specifically, program variables are assigned to sets of values instead of individual values.

The key challenge is modifying the program execution to handle sets of values. For example, if a Boolean variable $x$ is bound to the set of values $x \mapsto \{\text{True}, \text{False}\}$, and a conditional statement `if` $x$ `then` $p_{\text{true}}$ `else` $p_{\text{false}}$ that branches on $x$, then we effectively execute both branches $p_{\text{true}}$ and $p_{\text{false}}$ of the conditional; then, for each variable $y$ defined in these branches, we take the union of the values $v_{\text{true}}$ bound to $y$ in $p_{\text{true}}$ and $v_{\text{false}}$ in $p_{\text{false}}$, i.e., $y \mapsto v_{\text{true}} \cup v_{\text{false}}$. QUASAR includes a modified set of *conformal* rewrite rules that handle variables bound to sets of values in this way.

Because external functions are opaque to QUASAR, abstract versions of them must be provided. In the case of calls to neural models, such as `find`, the abstract version is provided by applying some conformal technique, such as returning the set of labels whose probability is above some threshold. For example, the object detector shown in the left of Figure 1 misses two objects (though in this case, it does not affect the final answer in Figure 2); the output of the conformal detector is shown on the right. In this case, the external call `image_patch.find("drink")` indicates whether each detection is definitely (green) or possibly (yellow) in the image; it represents the set of lists of patches

$$\{[\text{patch1}, \text{patch2}, \text{patch3}, \text{patch4}], [\text{patch2}, \text{patch3}, \text{patch4}],$$
$$[\text{patch1}, \text{patch2}, \text{patch4}], [\text{patch2}, \text{patch4}]\},$$

where the patches are ordered from left to right. Similarly, for each patch, the external call `patch.simple_query("Does this drink have alcohol?")` returns a prediction set that is a subset of $\{\text{"yes"}, \text{"no"}\}$. QUASAR overapproximates the true output; in this case, the program output is $\{\text{"yes"}\}$, i.e., there is definitely an alcoholic drink in the image.

The conformal guarantee says that, for some target fraction of the test dataset ("coverage"), the ground truth label will be contained in the predicted set of labels. While this can be trivially obtained by outputting the set of all labels, the sizes of sets should be kept as small as possible while satisfying the target coverage. To satisfy the desired coverage guarantee, we use a standard conformal prediction strategy. First, we optimize the thresholds for each individual model on a optimization set (Li et al., 2024b). Then, using a held-out calibration set, we jointly rescale these thresholds using a single scaling parameter $\tau \in \mathbb{R}$ chosen using conformal prediction to satisfy a desired coverage guarantee (Angelopoulos et al., 2022; Zhang et al., 2025). $\tau$ also determines the number of programs to generate, in order to handle uncertainty at the program level (Quach et al.).

```
                                                     ({77: '.find', 83: '.simple_query'},
                                                      ('def',
                                                       75,
                                                       ((76,),
                                                        ((('prim', 78, 'drink'),
                                                          ('call', (79,), 77, (76, 78)),
                                                          ('prim', 80, False),
drink_patches = image_patch.find("drink")             ('def',
found = False                                            81,
for drink_patch in drink_patches:                       ((89, 82),
    if drink_patch.simple_query("Does this have alcohol?"):   (((('prim', 84, 'Does this have alcohol?'),
        found = True                                      ('call', (85,), 83, (82, 84)),
return found                                              ('def', 86, ((), (((('prim', 87, True),), (87,)))),
                                                         ('def', 88, ((), ((), (89,))))),
                                                         ('call', (91,), 0, (85,)),
                                                         ('call', (92,), 91, (86, 88)),
                                                         ('call', (90,), 92, ())),
                                                        (90,)))),
                                                       ('call', (93,), 79, (80, 81)),
                                                       (93,)))))
```

(a)                                                      (b)

Figure 3: An example of the same agent code, in both Python (a) and raw QUASAR (b) forms.

## 3.5 GENERATING QUASAR CODE

For purposes of illustration, we have written example code with a syntax similar to Python. However, as shown in Figure 3, raw QUASAR code looks very different. A key challenge is that LLMs have never seen QUASAR code before, and we find that they struggle to generate it directly. Instead, our strategy is to have the LLM generate Python and then *transpile* this Python code to QUASAR. That is, the LLM generates the code in Figure 3a, we transpile it to the code in 3b, and then the QUASAR interpreter executes it. It is very challenging to transpile unrestricted Python to QUASAR, since this strategy would inherit all the challenges of making Python more performant, secure, and reliable. Furthermore, many practical agents do not use the unsupported language features of Python (e.g., classes and inheritance); intuitively, agents are trying to perform actions, not write complex software. Thus, our transpiler supports a restricted subset of Python carefully chosen to balance expressiveness and ease of transpilation. We consider three strategies for generating Python code in this subset:

- **Instruction:** It is often sufficient to instruct the LLM to do so in the system prompt.
- **Multi-turn:** It is also possible to feed error messages from our transpiler back to the LLM in a multi-turn loop. Importantly, this feedback is only based on static program information; in contrast, traditional multi-turn feedback based on Python interpreter errors requires running the code, which can result in undesirable side-effects from unsuccessfully executed code. However, a shortcoming of this approach is that calling the LLM multiple times can be slow.
- **SFT:** For smaller models which struggle to adhere to the allowed subset of Python, we can use supervised fine-tuning to improve adherence.

We provide details on the supported subset of Python and our transpilation strategy Appendix B.

## 4 EVALUATION

We evaluate two aspects of our approach. First, we show that generating QUASAR code via transpilation retains task performance comparable to the use of Python, and we show several strategies for improving transpilation success (Section 4.1). Second, we show that QUASAR is useful, offering improvements in several diverse regards: performance, with significant reductions in execution time (Section 4.2); security, with significant reductions in the number of user interactions required (Section 4.3); and reliability, with the conformal semantics achieving a target coverage rate (Section 4.4).

We evaluate on ViperGPT (Surís et al., 2023), a visual question answering agent approach, and CaMeL, a secure AI assistant. Given a natural language query about an image, ViperGPT first uses an LLM agent to generate a Python program that would answer that query when provided with an image. The Python program itself has access to various neural modules, including an object detector, a vision-language model, and an LLM. We apply the ViperGPT approach on 1000 tasks randomly sampled from GQA (Hudson & Manning, 2019), a dataset of questions about various day-to-day

| Approach | Execution | | Accuracy | |
|---|---|---|---|---|
| | **GQA** | **AD** | **GQA** | **AD** |
| Python | 99.6 | 76.7 | 71.8 | 64.6 |
| Ours | 99.9 | 84.4 | 73.1 | 65.7 |
| Multi-turn | 100.0 | 91.1 | 73.1 | 70.5 |
| Nano Base | 92 | – | 65 | – |
| Nano SFT | 99 | – | 71 | – |

Table 1: Comparison of different code generation approaches on GQA and AgentDojo (AD). "Execution" is the fraction of generated programs that execute successfully (i.e., no syntax or runtime errors); "Accuracy" is the fraction of successful programs that correctly output the ground truth label.

images. CaMeL performs tasks for users by using an LLM to generate Python code which itself may invoke "quarantined" LLMs to process data without risk of prompt injections. We apply the CaMeL approach to all four suites of the AgentDojo (Debenedetti et al., 2024) benchmark. All of our results use GPT-5 with default ("Medium") reasoning level unless otherwise stated.

## 4.1 Generation of Quasar Code

First, we compare ordinary, unrestricted Python generation ("Python") to Python generation restricted to the allowed subset ("Ours"), as well as using multi-turn feedback ("Multi-turn"). Finally, we consider fine-tuning small model (i.e., GPT-4.1-nano) using programs generated by a large model (i.e., GPT-5). We report results for both the base small model ("Nano Base") and the fine-tuned version ("Nano SFT"); for these results, we use 900 examples for training and report results on a held-out subset of 100 examples. For each approach, we consider the evaluation accuracy on the GQA and AgentDojo (AD) datasets—i.e., for what fraction of tasks does the generated program both execute without error ("Execution") and produce the correct result for the task ("Accuracy"). Execution errors can be due to the LLM failing to adhere to the allowed subset or due to runtime errors. We show results in Table 1. Restricting to the Python subset ("Ours") does not degrade accuracy for either dataset. Furthermore, multi-turn feedback improves accuracy for AgentDojo ("Multi-turn"); for GQA, transpilation is almost always successful so we see no improvement. Finally, SFT improves accuracy for GQA; the AgentDojo dataset contains only 93 tasks, which is not enough for SFT.

## 4.2 Performance

To evaluate the performance improvements, we consider pairs of QUASAR programs and the Python programs that they were transpiled from, ensuring that the programs have the same input-output behavior. We also control for the time that each external call takes to execute by recording every external call that a program makes and what its result and running time are. Then, we replay this recording on both the Python and QUASAR versions of the program and record the total execution time of each. The running times of these program pairs are shown in Figure 4a. On the GQA dataset, QUASAR reduces running time by $18\% \pm 23$ (mean $\pm$ stddev). This large variance is because only $49\%$ of tasks are parallelizable. Among those, the running time is cut by $37\% \pm 19$. On AgentDojo, the overall speedup is $49\% \pm 32$, with $67\%$ improvable and a speedup of $56\% \pm 22$ on those.

## 4.3 Security

We evaluate the security improvements in terms of the reduction in the number of user interactions required to approve all external calls made by the program. As in Section 4.2, we consider pairs of equivalent Python and QUASAR programs, i.e., that make exactly the same external calls. We compare the number of user approvals required if the external calls are approved one at a time versus if they are approved in batches (i.e., QUASAR executes as much internally as possible before asking the user to approve). We show results in Figure 4c. On GQA, QUASAR reduces the number of user interactions by $26\% \pm 29$. The large variance is because only $50\%$ of tasks offer batching of approvals; among those, the interaction count is more than cut in half, by $53\% \pm 17$. On AgentDojo, the overall reduction is $25\% \pm 26$, with $56\%$ improvable and a reduction of $55\% \pm 19$ on those.

| Dataset | Performance | | | Security | | | Reliability | | |
|---|---|---|---|---|---|---|---|---|---|
| | Overall Speedup | Fraction Improvable | Improvable Speedup | Overall Reduction | Fraction Improvable | Improvable Reduction | Target Error | Empirical Error | Fraction Uncertain |
| GQA | $18\% \pm 23$ | 49% | $37\% \pm 19$ | $26\% \pm 29$ | 50% | $53\% \pm 17$ | 10% | $7.9\% \pm 2.8$ | $58.2\% \pm 8.3$ |
| AD | $49\% \pm 32$ | 67% | $56\% \pm 22$ | $25\% \pm 26$ | 56% | $44\% \pm 19$ | 10% | $4.0\% \pm 3.6$ | $92.6\% \pm 5.1$ |

Table 2: An overview of the improvements (mean $\pm$ stddev) provided by QUASAR.

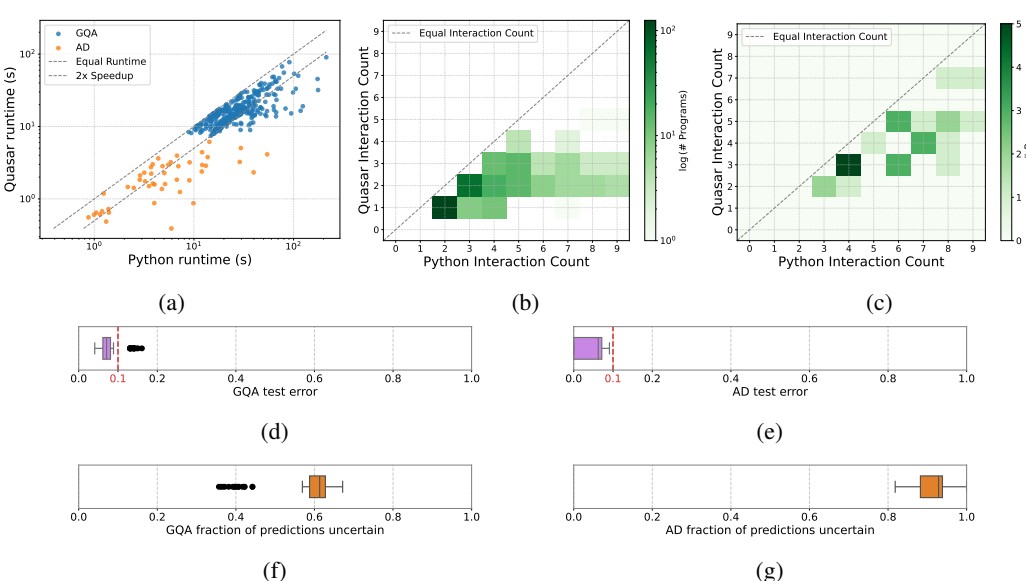

(a)                                    (b)                                    (c)

(d)                                                        (e)

(f)                                                        (g)

Figure 4: Python vs QUASAR running time for the improvable tasks (a). Python vs QUASAR user interactions required for the improvable GQA tasks (b) and AgentDojo tasks (c). Using the conformal semantics and targeting 0.1 coverage, the distribution of coverage for GQA (d) and AgentDojo (e), and the distribution of fraction of "uncertain" predictions for GQA (g) and AgentDojo (g) for 100 different validation/test splits. Note that the high rate of uncertain predictions in AgentDojo is due to our stringent criterion (i.e., we count any set size bigger than one as uncertain).

## 4.4 RELIABILITY

We evaluate reliability by showing how the conformal semantics can achieve a target error rate of 0.1 on a test set. Using the same dataset of QUASAR programs, we evaluated using the conformal semantics with several different threshold values, which produce progressively larger output sets for each program. We divided the dataset 100 times into validation/test splits. For each split, we chose the largest threshold (and thus smallest prediction sets) where the validation error was less than 0.1, and then we computed the test error with that threshold. The distribution of these test errors is shown for GQA in Figure 4d, with mean coverage $7.9\% \pm 2.8$ and for AgentDojo in Figure 4e, with mean coverage $4.0\% \pm 3.6$. Because the domain of labels varies based on task (e.g., yes/no, color, object, etc), instead of measuring the size of prediction sets we measure certainty—i.e., the model is certain if the prediction set is size 1, and otherwise it is uncertain. We consider the fraction of tasks on which the model is uncertain. The distribution of such uncertainty rates in GQA is shown in Figure 4f, with mean $58.2\% \pm 8.3$, and for AgentDojo in Figure 4g, with mean $92.6\% \pm 5.1$.

## 5 CONCLUSION

We have presented QUASAR, a language for code actions by LLM agents. Leveraging LLMs proficiency with Python, we transpile from a subset of Python into QUASAR. Compared to Python, QUASAR offers several key benefits in terms of performance (via automatic parallelization), security (by dynamically asking the user for approval of batches of external calls), and reliability (by supporting offering conformal execution semantics for programs).

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

$$P ::= stmt_1; \ldots; stmt_n; \text{return } x$$
$$stmt ::= x \leftarrow op$$
$$op ::= \text{prim } c$$
$$\mid x$$
$$\mid (x_1, \ldots, x_n)$$
$$\mid f\ x$$
$$\mid \text{proj } i\ x$$
$$\mid \text{fold } w\ x\ block$$
$$\mid \text{if } x\ block_1\ block_2$$
$$\mid ?S$$
$$block ::= \{x \Rightarrow P\}$$
$$value ::= c \mid (value_1, \ldots, value_n)$$

Figure 5: The grammar defining programs in QUASAR.

## A    FULL QUASAR LANGUAGE

As described in Section 3, QUASAR executes programs by transforming them with rewrite rules until they reach a result. The syntax of programs is given in Figure 5. A program consists of a sequence of statements, where each statement defines some variable $(x, y, \ldots)$ to be the result of some operation $(op)$. Variables are assumed to be defined exactly once (i.e., they are unique and do not shadow each other). An operation can be, in order: a primitive value $c$ (where $c$ ranges over Python values, such as `True`, `5`, or `"foo"`); another variable $x$; a tuple of variables $x_i$, the result of calling an external function $f$ (from the set $\mathcal{F}_{ext}$) with argument $x$; the result of projecting out the $i$-th component from a tuple $x$; the result of folding over a list $w$ with initial accumulator $x$ and fold body $block$; an if expression on condition $x$ with then-case $block_1$ and else-case $block_2$; or the result of some pending external call $S$. A block is a program, but which may additionally have some parameter $x$ (in particular, so that the body of a fold can take the previous accumulator and the current list item as arguments). A value is either a Python object or a (possibly nested) tuple of Python objects—it does not directly occur in programs, but is used in the semantics.

The interpreter state at any time is simply a program $P$ and a set $E$ of dispatched external calls. The semantics consist of rewrite rules $R \in \mathcal{R}$, which transform one execution state to another, written $P, E \xrightarrow{R} P', E'$. Many rules do not affect $E$, and so are simply written as $P \xrightarrow{R} P'$.

The rewrite rules are given in Figure 6. The rule "alias" removes a statement $y \leftarrow x$, replacing it with nothing, but renaming all occurences of $y$ in the program to $x$; "proj" replaces a projection operator, if the variable $x$ is known to be a tuple $(x_1, \ldots, x_n)$, with the $i$-th element; for if statements, when the condition $x$ is the primitive `True` ("if-t"), then the statement is replaced by a copy of $block_1$ (copying ensures that variables are unique; since blocks in if statements do not require parameters, $w$ is bound to an empty tuple); if the condition is `False` ("if-f"), then the same is done for $block_2$; "fold" applies $block$ to each element of the list $w$, with $x$ being the initial accumulator and $y$ being the final one, and $z_i$ being the $i$-th intermediate accumulator; "disp" replaces an external call to a function $f$ when the argument $x$ has a value $value$ (i.e., it is a primitive or a tuple of primitives, which $\text{value}(T, x)$ computes) with a placeholder $S$, begins executing the function $f$, and updates the execution set $E$; "ext" applies when an external function has finished executing—and so the execution set $E$ contains a result in place of $\varnothing$—and replaces the placeholder with the result. In Section 3, we simplified $S$ in the execution set to just be the external call statement itself, whereas here it is an identifier for the spawned task.

## B    TRANSPILATION

QUASAR is functional, while Python supports imperative programming. Thus in QUASAR, variables cannot be changed once they have been defined. Being functional makes supporting parallel, partial,

$$\frac{}{T[y \leftarrow x] \rightarrow \mathrm{rename}(T[[\varnothing]], y, x)}\text{(alias)} \qquad \frac{(x \leftarrow (x_1, \ldots, x_n)) \in T}{T[y \leftarrow \mathrm{proj}\ i\ x] \rightarrow T[[y \leftarrow x_i]]}\text{(proj)}$$

$$\frac{(x \leftarrow \mathrm{prim}\ \mathtt{True}) \in T \qquad \{w \Rightarrow stmts; \mathrm{return}\ z\} = \mathrm{copy}(block_1)}{T[y \leftarrow \mathrm{if}\ x\ block_1\ block_2] \rightarrow T[[w \leftarrow (); stmts; y \leftarrow z]]}\text{(if-t)}$$

$$\frac{(x \leftarrow \mathrm{prim}\ \mathtt{False}) \in T \qquad \{w \Rightarrow stmts; \mathrm{return}\ z\} = \mathrm{copy}(block_2)}{T[y \leftarrow \mathrm{if}\ x\ block_1\ block_2] \rightarrow T[[w \leftarrow (); stmts; y \leftarrow z]]}\text{(if-f)}$$

$$\frac{\begin{array}{c}(w \leftarrow \mathrm{prim}\ [c_1, \ldots, c_n]) \in T \\ \forall i.\{y_i \Rightarrow stmts_i; \mathrm{return}\ z_i\} = \mathrm{copy}(block) \qquad stmts_i' = (w_i \leftarrow \mathrm{prim}\ c_i; y_i \leftarrow (z_{i-1}, w_i); stmts_i)\end{array}}{T[y \leftarrow \mathrm{fold}\ w\ x\ block] \rightarrow T[[z_0 \leftarrow x; stmts_1'; \ldots; stmts_n'; y \leftarrow z_n]]}\text{(fold)}$$

$$\frac{\mathrm{value}(T, x) = value \qquad S = \mathrm{spawn}(f, value)}{T[y \leftarrow f\ x], E \rightarrow T[[y \leftarrow ?S]], E \cup \{(S, \varnothing)\}}\text{(disp)}$$

$$\frac{term = (stmts; \mathrm{return}\ x)}{T[y \leftarrow ?S], E \cup \{(S, term)\} \rightarrow T[[stmts; y \leftarrow x]], E}\text{(ext)}$$

(a)

$$\frac{(x \leftarrow \mathrm{prim}\ c) \in T}{\mathrm{value}(T, x) = c} \qquad \frac{(x \leftarrow (x_1, \ldots, x_n)) \in T \qquad \forall i.\,\mathrm{value}(T, x_i) = value_i}{\mathrm{value}(T, x) = (value_1, \ldots, value_n)}$$

(b)

Figure 6: The rewrite rules of the semantics of QUASAR (a), and the formal definition of the value function used by the "disp" rule (b). $T[stmt]$ means a program $P$ with some statement $stmt$ in it; $T[[stmts]]$ means that the statement $stmt$ was replaced by the list of statements $stmts$. For each rule, the $P, E \rightarrow P', E'$ below the line is an allowed rewrite, subject to all of the conditions above the line. If $E$ is not modified by a rule, we omit it for concision. $stmt \in T$ means that $stmt$ is in the list of statements of $T$.

```
found = False
cond = drink_patch.simple_query("...")
if cond:
    found = True
```

⤳

```
found0 = False
cond = drink_patch.simple_query("...")
def then_case():
    found1 = True
    return found1
def else_case():
    return found0
found2 = then_case() if cond else else_case()
```

Figure 7: An illustration of the translation of "if" statements, shown in Python syntax. Before, with imperative updates in "if" statements (a). After, with no imperative updates and a functional conditional operation (b).

and conformal execution possible, but agents generate Python code with imperative variable updates to local variables. Handling updates in "straight-line" code (without control-flow structures like `if` and `for`) is straightforward. However, updates inside of control-flow structures, which thus may or may not happen, are more challenging. In the example from Figure 1, `found` may be updated inside of the loop.

Currently, QUASAR supports the subset of Python that uses function calls, local variable assignments, and `if`, `for`, and `while` control-flow constructs. It does not support early returns from loops (i.e., `break`, `continue`, or `return` inside of a loop). To support imperative control-flow structures in Python, we convert them into a functional form. `if` statements in Python are transformed as shown in Figure 7, where variables that might be updated by a statement-level conditional are instead returned from an expression-level conditional. A similar translation is done from imperative `for` loops to functional fold operations: variables that might be updated by the `for` loop are instead passed as the fold accumulator (in Python, this fold operation is called `reduce`).

$$op ::= \ldots$$
$$| \text{ absprim } \{c_1, \ldots, c_n\}$$
$$| \text{ abslist } [(c_1, b_1), \ldots, (c_n, b_n)]$$
$$| \text{ join } \{x_1, \ldots, x_n\}$$
$$absvalue ::= \{c_1, \ldots, c_n\} \mid (absvalue_1, \ldots, absvalue_n)$$

Figure 8: The additional operations in the grammar of QUASAR to support conformal evaluation.

$$\frac{y \leftarrow \text{join } y_1, \ldots, y_m}{T[x \leftarrow \text{join } x_1, \ldots, x_n, y] \rightarrow T[[x \leftarrow \text{join } x_1, \ldots, x_n, y_1, \ldots, y_m]]} \text{(join-join)}$$

$$\frac{(x_i \leftarrow (w_{i,1}, \ldots, w_{i,m})) \in T \qquad stmts_j = (y_j \leftarrow \text{join } w_{1,j}, \ldots, w_{n,j})}{T[x \leftarrow \text{join } x_1, \ldots, x_n] \rightarrow T[[stmts_1; \ldots; stmts_n; x \leftarrow (y_1, \ldots, y_m)]]} \text{(join-tuple)}$$

$$\frac{(x_i \leftarrow \text{prim } c_i) \in T}{T[x \leftarrow \text{join } x_1, \ldots, x_n] \rightarrow T[[x \leftarrow \text{absprim } \{c_1, \ldots, c_n\}]]} \text{(join-prim)}$$

$$\frac{(x \leftarrow \text{absprim } \{\texttt{True}, \texttt{False}\}) \in T}{\{w_1 \Rightarrow stmts_1; \text{return } z_1\} = \text{freshen}(block_1) \qquad \{w_2 \Rightarrow stmts_2; \text{return } z_2\} = \text{freshen}(block_2)}{T[y \leftarrow \text{if } x \ block_1 \ block_2] \rightarrow T[[w_1 \leftarrow (); stmts_1; w_2 \leftarrow (); stmts_2; y \leftarrow \text{join } z_1, z_2]]} \text{(if-tf)}$$

$$\frac{(w \leftarrow \text{abslist } [(c_1, b_1), \ldots, (c_n, b_n)]) \in T}{\forall i.\{y_i \Rightarrow stmts_i; \text{return } z_i\} = \text{freshen}(block) \qquad stmts_i' = (w_i \leftarrow \text{prim } c_i; y_i \leftarrow (z_{i-1}, w_i); stmts_i)}{stmts_i'' = \text{if } b_i \text{ then } stmts_i' \text{ else } (stmts_i'; z_i \leftarrow \text{join } z_i, z_{i-1})}{T[y \leftarrow \text{fold } w \ x \ block] \rightarrow T[[z_0 \leftarrow x; stmts_1''; \ldots; stmts_n''; y \leftarrow z_n]]} \text{(fold-abs)}$$

Figure 9: The additional rewrite rules in the semantics of QUASAR to support conformal evaluation.

## C  CONFORMAL SEMANTICS FOR QUASAR

In order to support conformal evaluation, QUASAR must be extended to support sets of values. The syntax has three additional operations, as shown in Figure 8. In order: abstract primitives represent one of a set of Python values, $c_i$; abstract lists represent a list where some of the elements may be uncertain: if $b_i$ is False the element $c_i$ may or may not be in the list, whereas if $b_i$ is False, then $c_i$ is definitely in the list; and a join operation, which combines two computations into a set. Join is distinct from an abstract set, since in the latter the values must be known, whereas in the former they may not yet be computed.

The semantics also contains additional rules in order to support these new operations, as shown in Figure 9. The rule "join-join" applies when $x$ is the join of variables, and one of them, $y$, is itself a join, in which case they can be flattened to a single join; "join-tuple" applies when $x$ is the join of $n$ tuples of identical length $m$, in which case it becomes the tuple of joins of the respective components; "join-prim" applies when $x$ is the join of $n$ primitives $c_i$, in which case it becomes an abstract set of those values; "if-tf" applies when the condition of an if statement is the abstract set of both True and False, in which case both branches are taken, resulting in $z_1$ and $z_2$, which are joined to produce $y$; "fold-abs" applies when folding over an abstract list, in which case a copy of $block$ is made for each element of the list, however if a list element is uncertain ($b_i = $ False), then the resulting accumulator $z_i$ is joined with $z_{i-1}$ to capture both the case when $c_i$ is and is not in the list.

