# OpenReview forum: "A Fast, Reliable, and Secure Programming Language for LLM Agents with Code Actions"
_ICLR.cc/2026/Conference — Submitted to ICLR 2026_

### Official Review · Reviewer_MYLk · 2025-10-15

**Soundness:** 2
**Presentation:** 2
**Contribution:** 2
**Rating:** 6
**Confidence:** 1

**Summary:**

This paper falls outside my area of expertise.  I'm unable to assess this paper.

**Strengths:**

N/A

**Weaknesses:**

N/A

**Questions:**

N/A

---

### Official Review · Reviewer_JjBk · 2025-11-01

**Soundness:** 3
**Presentation:** 3
**Contribution:** 3
**Rating:** 4
**Confidence:** 4

**Summary:**

This paper introduces QUASAR, a novel programming language designed to improve the performance, security, and reliability of LLM-based agents. QUASAR achieves this by separating internal computations from external side effects, supporting parallel execution, and utilizing conformal semantics for uncertainty quantification. The language is implemented through a workflow where LLMs generate Python code, which is then transpiled to QUASAR. Experimental results show that QUASAR improves execution speed (up to 56% faster), reduces user interaction for security validation, and maintains high reliability with a target error rate of 0.1.

**Strengths:**

- The paper effectively identifies challenges in LLM-based agents which write Python code to invoke tool APIs, and presents a practical solution through QUASAR.
- The "internal computation - external side effects separation" architecture and the introduction of conformal semantics are novel and offer significant advantages in performance, security, and reliability.
- Experiments on real-world agents like ViperGPT and CaMeL, covering performance, security, and reliability, demonstrate the practical benefits of QUASAR.

**Weaknesses:**

- Lack of Detailed Technical Explanation: The paper lacks in-depth descriptions of key components like QUASAR’s rewrite rules, Python subset syntax, and transpiler implementation, which could impact reproducibility and understanding.
- Flexibility Concerns in Tool-Calling Scenarios: While QUASAR improves upon Python in certain areas, there is a concern about whether it can maintain the same flexibility as Python in all tool-calling scenarios. Python’s ecosystem is rich with libraries that facilitate diverse use cases (e.g., system administration tasks, network programming, data processing). It’s unclear if QUASAR can handle such diverse scenarios with the same ease and flexibility, particularly in more dynamic, real-time applications where Python’s built-in flexibility is often crucial. A clearer discussion on this aspect and how QUASAR addresses such scenarios, if at all, would be valuable.

**Questions:**

- What optimizations would you suggest for fine-tuning on small datasets like AgentDojo? Are techniques like transfer learning or data augmentation being considered to improve performance on such tasks?
- How does QUASAR address tasks that cannot be parallelized due to dependencies? Could you provide more insight into the characteristics of tasks that hinder parallel execution?
- Could QUASAR be considered more of a specialized Python interpreter rather than a new programming language? How does it differ from existing solutions such as parallel-execution Python interpreters (PyPy, Cython) or security frameworks (e.g., Sandboxed Python), which already provide performance improvements for Python code? What makes QUASAR’s approach more beneficial than these well-established, mature solutions?

---

> ### Author Response · Authors · 2025-11-21
>
> ## Lack of Detailed Technical Explanation: The paper lacks in-depth descriptions of key components like QUASAR’s rewrite rules, Python subset syntax, and transpiler implementation, which could impact reproducibility and understanding.
> The grammar and rewrite rules of Quasar are provided in Appendix A (Figures 5 and 6). Appendix B describes the Python subset syntax and the transpiler. We are happy to elaborate on any aspects of the formalism. Additionally, we plan to release our implementation for reproducibility.
>
> ## Flexibility Concerns in Tool-Calling Scenarios: While QUASAR improves upon Python in certain areas, there is a concern about whether it can maintain the same flexibility as Python in all tool-calling scenarios. Python’s ecosystem is rich with libraries that facilitate diverse use cases (e.g., system administration tasks, network programming, data processing). It’s unclear if QUASAR can handle such diverse scenarios with the same ease and flexibility, particularly in more dynamic, real-time applications where Python’s built-in flexibility is often crucial. A clearer discussion on this aspect and how QUASAR addresses such scenarios, if at all, would be valuable.
> Quasar is designed to make external calls out to arbitrary Python code, and as such, can leverage the existing Python ecosystem. This won't automatically yield the benefits of Quasar (e.g. parallelization), but for commonly used (e.g. LLM calls, numpy, or filesystem APIs) and relevant APIs small annotations can be provided, as we have done for the APIs we use. In future work, we would like to explore automatically generating these annotations, e.g. by using LLMs to read API documentation to determine e.g. parallelizability. Similar annotations are used to determine safety of calls (whether they need user approval) and to provide conformal versions of calls (e.g. for language and vision models that should return sets of labels).
>
> ## What optimizations would you suggest for fine-tuning on small datasets like AgentDojo? Are techniques like transfer learning or data augmentation being considered to improve performance on such tasks?
> In future work, we envision finetuning on a broader range of tasks, to train a general model to that outputs in the Python subset, but building such a dataset is beyond the scope of our paper.
>
> ## How does QUASAR address tasks that cannot be parallelized due to dependencies? Could you provide more insight into the characteristics of tasks that hinder parallel execution?
> Parallel execution is blocked by data-dependencies between external calls. E.g. `x = llm("foo"); y = llm(x)` has `x` as a data-dependency, whereas `x = llm("foo"); y = llm("bar")` has no data-dependency. In cases with data-depenencies, Quasar executes tasks sequentially (however, it may execute other, independent tasks in the program in parallel). However, it's not clear that it is even possible to execute tasks with data-dependencies in parallel.
>
> On the GQA and AgentDojo datasets, it is common for agents to make an LLM or vision model call to get a list of concepts or objects, and then to loop over that list and do an LLM call for each item. In these cases, the loop iterations are mostly independent and can be parallelized, but the initial call that returns a list must be done first, and cannot be parallelized.
>
> ## Could QUASAR be considered more of a specialized Python interpreter rather than a new programming language? How does it differ from existing solutions such as parallel-execution Python interpreters (PyPy, Cython) or security frameworks (e.g., Sandboxed Python), which already provide performance improvements for Python code? What makes QUASAR’s approach more beneficial than these well-established, mature solutions?
> We consider QUASAR to be a different programming language because its semantics are so fundamentally different from that of Python. Unlike existing interpreters and security frameworks, we do not believe it is feasible to implement QUASAR semantics directly on Python ASTs, and we view transpilation as a critical step for enabling the benefits of QUASAR.
>
> PyPy and Cython focus on speeding up execution of Python code, but they do so by optimizing the code, not by executing external calls out-of-order and in parallel (for code actions, the execution bottleneck is almost never in the code, but instead waiting for external APIs).
>
> Sandboxing approaches such as Blaxel and Docker restrict the agent from accessing local resources (e.g. files) without approval. However, because the code actions are executing sequentially, the user must provide approval sequentially. Only through Quasar's out-of-order execution, which no existing system does, are we able to collect many external calls requiring approval and present them to the user simultaneously.

---

> ### Author Response · Authors · 2025-12-01
>
> We have added to Quasar's evaluation the BrowseComp-Plus deep research benchmark. The agent we used has been introduced in prior work as Recursive Language Models, where rather than providing an LLM with all document in the context at once, which can exceed the context size or cause context rot, the LLM agent programatically manipulates the context (as a list of documents in a Python interpreter), and is able to call sub-LLMs.
>
> We implemented this agent using Quasar rather than Python. For time and cost reasons (due to the large number of sub-LLM calls required by each task), we did a preliminary evaluation on 80 randomly selected tasks (out of 800 in the full dataset) and used GPT-5 nano.
>
> We see similar results as on our other two benchmarks: the rate of successful program executions is similar between Python (99%) and Quasar (100%), as is the task success rate for Python (49%) and Quasar (50%). The success rates would likely be higher with larger or finetuned models. Quasar yields large speedups: across the entire dataset there is a 63% +- 147 speedup, and for the set of programs that improve, there is a 92% +- 8 speedup. Across the entire dataset, there is a 97% +- 1 reduction in the number of user approvals required.
>
> We will add an evaluation on the full dataset to our paper once it is completed.

---

### Official Review · Reviewer_M2ub · 2025-11-02

**Soundness:** 3
**Presentation:** 3
**Contribution:** 3
**Rating:** 6
**Confidence:** 2

**Summary:**

The paper introduces QUASAR, a new programming language designed to make LLM-driven code execution faster, safer, and statistically reliable. It combines a pure functional core, explicit side-effect isolation, automatic parallelization, and conformal prediction–based uncertainty propagation. The work is ambitious and conceptually motivated, aiming to establish a formal, language-level foundation for trustworthy agent behavior. Broadening experiments and addressing realistic LLM integration would make it stronger in practice and more convincing.

**Strengths:**

- Designing an LLM-native programming language for code generation action is innovative and promising.
- QUASAR introduces a pure functional core that separates computation from side effects.
This separation allows deterministic execution, simplifies formal reasoning, and makes program behavior easier to verify and audit.
- The runtime system can automatically detect independent external calls and execute them concurrently.
Experiments show up to 56% reduction in total execution time, demonstrating concrete performance gains compared to sequential baselines.
- QUASAR enforces strict external-call isolation through explicit user approval.
It introduces a batch-approval mechanism that reduces the number of user interactions by over 50%, balancing usability and safety while preventing unverified API calls.

**Weaknesses:**

- **Narrow evaluation scope:**
The experiments are confined to small, synthetic benchmarks (GQA and AgentDojo). These tasks are short and prestructured, which limits the external validity of the claims. There is no evaluation in complex or dynamic environments that real LLM agents operate in.
- **Limited language expressiveness:**
QUASAR only supports a very restricted subset of Python (functions, variables, simple control flow). It does not handle classes, exceptions, pattern matching, or early returns. This simplicity makes formal analysis easier but severely limits applicability to realistic agent workflows that depend on richer language features.
- **Unclear LLM integration strategy:**
The paper proposes transpiling from a restricted Python subset but does not explain how LLMs are constrained to generate only this subset. There is no discussion of prompting or fine-tuning when the model produces invalid constructs. This leaves a major usability gap between theory and practice.

**Questions:**

How are LLMs guided or constrained to produce valid Python subsets that can be reliably transpiled into QUASAR, and what is the success rate of this process in practice?

---

> ### Author Response · Authors · 2025-11-21
>
> ## Narrow evaluation scope: The experiments are confined to small, synthetic benchmarks (GQA and AgentDojo). These tasks are short and prestructured, which limits the external validity of the claims. There is no evaluation in complex or dynamic environments that real LLM agents operate in.
> First, while we agree that these tasks are simple, we disagree that they are not realistic; while agents certainly operate in more complex environments, we believe AgentDojo tasks are representative of simpler use cases that still have significant practical value. Furthermore, we note that being prestructured is not a limitation of our approach, we simply use this approach following the existing ViperGPT and CaMeL strategies. More broadly, our tasks are in line with evaluations we have seen in recent work on agents with code actions.
>
> ## Limited language expressiveness: QUASAR only supports a very restricted subset of Python (functions, variables, simple control flow). It does not handle classes, exceptions, pattern matching, or early returns. This simplicity makes formal analysis easier but severely limits applicability to realistic agent workflows that depend on richer language features.
> We emphasize that Quasar is a programming language for code actions, not for software development. While advanced language features like exceptions and classes are undoubtedly useful for software development, and thus software development agents, we disagree that they are important for code actions, which are a fundamentally different use-case, and typically involve relatively short snippets of self-contained code.
>
> On the other hand, early return/break/continue are constructs that commonly occur in code actions for existing LLM agents. However, these constructs are not critical for expressiveness (i.e. there is always a rewriting of the loop that does not have early return/break/continue but produces the same result). Moreover, our experiments demonstrate that code without early return/break/continue is able to achieve the same performance in practice (see Figure 1, "Python" vs "Ours").
>
> Finally, the lack of these features is not a fundamental limitation of our approach, and they could be added to both the Quasar language and the compiler with additional engineering effort, should they become necessary.
>
> ## Unclear LLM integration strategy: The paper proposes transpiling from a restricted Python subset but does not explain how LLMs are constrained to generate only this subset. There is no discussion of prompting or fine-tuning when the model produces invalid constructs. This leaves a major usability gap between theory and practice.
> We took three approaches to generating code in this subset (see Table 1): "Ours" merely included the constraints in the prompt instructions, producing successfully compiling + executing code 99.9% of the time (GQA) and 84.4% of the time (AgentDojo). "Multi-turn" feeds transpiler errors back into the LLM and asks it to regenerate; this boosts the success rate on AgentDojo to 91.1%. We also considered supervised finetuning of a smaller model (GPT-4.1-nano instead of GPT-5); the smaller base model only gets 92% execution success on AgentDojo, but when finetuned this increases to 99%.

---

> ### Author Response · Authors · 2025-12-01
>
> We have added to Quasar's evaluation the BrowseComp-Plus deep research benchmark. The agent we used has been introduced in prior work as Recursive Language Models, where rather than providing an LLM with all document in the context at once, which can exceed the context size or cause context rot, the LLM agent programatically manipulates the context (as a list of documents in a Python interpreter), and is able to call sub-LLMs.
>
> We implemented this agent using Quasar rather than Python. For time and cost reasons (due to the large number of sub-LLM calls required by each task), we did a preliminary evaluation on 80 randomly selected tasks (out of 800 in the full dataset) and used GPT-5 nano.
>
> We see similar results as on our other two benchmarks: the rate of successful program executions is similar between Python (99%) and Quasar (100%), as is the task success rate for Python (49%) and Quasar (50%). The success rates would likely be higher with larger or finetuned models. Quasar yields large speedups: across the entire dataset there is a 63% +- 147 speedup, and for the set of programs that improve, there is a 92% +- 8 speedup. Across the entire dataset, there is a 97% +- 1 reduction in the number of user approvals required.
>
> We will add an evaluation on the full dataset to our paper once it is completed.

---

### Official Review · Reviewer_FyqM · 2025-11-03

**Soundness:** 4
**Presentation:** 3
**Contribution:** 3
**Rating:** 6
**Confidence:** 2

**Summary:**

This paper introduces QUASAR, a novel programming language designed specifically for LLM agents that use code actions. Unlike Python—which is the standard medium for LLM-generated code—QUASAR provides built-in mechanisms for performance optimization (via automatic parallelization), security (via dynamic access control and user approval of external calls), and reliability (via conformal semantics for uncertainty quantification).

**Strengths:**

+ The rewrite-rule semantics and external call dispatch mechanism are rigorously formalized.

+ The ability to propagate model uncertainty at the program level is a novel contribution that could inspire future work on trustworthy agent execution.

+ The use of a Python subset and a transpiler ensures backward compatibility with current LLMs, addressing real-world deployability concerns (without performance degradation).

**Weaknesses:**

- The paper does not specify how QUASAR manages external call failures, exceptions, or thread-level errors. For example, what happens if an external API call fails, times out, or returns an invalid response? Is the failure propagated, retried, or absorbed?

- While QUASAR executes external calls “as soon as all their arguments are available,” it is not clear whether “futures” or deferred results are explicitly represented in the language. How does the interpreter manage dependencies among pending external calls or enforce order when results are reused?

**Questions:**

It wasn’t clear to me what the sentence “There is only one external rule Rext = {Rext}. This rule is designed to enable calls to external functions f ∈ Fext” means in practice. Does this imply that all side-effecting operations (e.g., API calls, LLM queries) are handled uniformly through this single rewrite rule? How does QUASAR distinguish between different external APIs at runtime?

Could QUASAR’s transpilation strategy be generalized to other host languages (e.g., typescript) or is it fundamentally tied to Python’s semantics?

---

> ### Author Response · Authors · 2025-11-21
>
> ## The paper does not specify how QUASAR manages external call failures, exceptions, or thread-level errors. For example, what happens if an external API call fails, times out, or returns an invalid response? Is the failure propagated, retried, or absorbed?
> API calls typically fail either due to transient issues, such as timeouts, or fatal issues, such as authentication failures. A key feature of Qusar is the ability to transparently retry in cases of transient errors, without blocking execution of the rest of the program. Currently, fatal errors end termination of the whole program, as Quasar does not have exception-handling (e.g. try-catch). However, such constructs are uncommon in agent-generated code, since typically the code block just fails, and then the agent itself decides what to do with the error. There is no fundamental limitation to adding exception handling in future work. We will clarify this.
>
> ## While QUASAR executes external calls “as soon as all their arguments are available,” it is not clear whether “futures” or deferred results are explicitly represented in the language. How does the interpreter manage dependencies among pending external calls or enforce order when results are reused?
> Futures are not exposed as a language construct to the programmer/LLM agent. Instead, though Quasar does not actually use futures internally, the way to think about its execution model is as if *every* program variable is automatically a future. The only time these "futures" are "awaited" is when they are arguments to an external call; an external call will be executed once all of its arguments have been "awaited". Please let us know if this answers your question.
>
> ## It wasn’t clear to me what the sentence “There is only one external rule Rext = {Rext}. This rule is designed to enable calls to external functions f ∈ Fext” means in practice. Does this imply that all side-effecting operations (e.g., API calls, LLM queries) are handled uniformly through this single rewrite rule? How does QUASAR distinguish between different external APIs at runtime?
> Yes, all side-effecting operations are handled through Rext. However, Rext is able to distinguish between external APIs because it can inspect the line making the function call and see which function is being called.
>
> ## Could QUASAR’s transpilation strategy be generalized to other host languages (e.g., typescript) or is it fundamentally tied to Python’s semantics?
> The transpilation strategy is not fundamentally tied to Python's semantics and can be applied to other languages.

---

> > ### Author Response · Authors · 2025-12-01
> >
> > We have added to Quasar's evaluation the BrowseComp-Plus deep research benchmark. The agent we used has been introduced in prior work as Recursive Language Models, where rather than providing an LLM with all document in the context at once, which can exceed the context size or cause context rot, the LLM agent programatically manipulates the context (as a list of documents in a Python interpreter), and is able to call sub-LLMs.
> >
> > We implemented this agent using Quasar rather than Python. For time and cost reasons (due to the large number of sub-LLM calls required by each task), we did a preliminary evaluation on 80 randomly selected tasks (out of 800 in the full dataset) and used GPT-5 nano.
> >
> > We see similar results as on our other two benchmarks: the rate of successful program executions is similar between Python (99%) and Quasar (100%), as is the task success rate for Python (49%) and Quasar (50%). The success rates would likely be higher with larger or finetuned models. Quasar yields large speedups: across the entire dataset there is a 63% +- 147 speedup, and for the set of programs that improve, there is a 92% +- 8 speedup. Across the entire dataset, there is a 97% +- 1 reduction in the number of user approvals required.
> >
> > We will add an evaluation on the full dataset to our paper once it is completed.

---

### Meta-Review · Area_Chair_a1iG · 2025-12-23

**Summary:**

This paper introduces QUASAR, a programming language for LLM agents that separates pure functional computation from external side effects to enable automatic parallelization, dynamic access control with batch user approvals, and conformal semantics for uncertainty quantification. The work demonstrates execution speedups up to 56% and over 50% reduction in user approval interactions on GQA and AgentDojo benchmarks. The strengths include novel language-level mechanisms for propagating model uncertainty, rigorous formalization of rewrite-rule semantics, successful transpilation from a Python subset maintaining backward compatibility, and concrete performance gains. The weaknesses center on limited evaluation scope using small, prestructured benchmarks that may not represent complex real-world agent workflows, restricted language expressiveness lacking classes, exceptions, and early returns that could limit practical applicability, incomplete specification of error handling and futures representation, and insufficient empirical validation of flexibility across diverse tool-calling scenarios. The decision to reject is based primarily on the narrow experimental validation that does not convincingly demonstrate generalizability to realistic agent deployments, coupled with borderline reviewer enthusiasm where all positive scores sit exactly at the marginal threshold.

**Reviewer Concerns:**

The authors provided technically sound responses to most reviewer concerns. For FyqM, they clarified that transient errors are transparently retried while fatal errors terminate execution, explained that futures are implicit on all variables rather than exposed constructs, and confirmed the single external rule distinguishes APIs through inspection. For M2ub, they defended benchmark realism for simpler practical use cases, argued that advanced language features are unnecessary for code actions versus software development, and provided concrete LLM integration success rates ranging from 84.4% to 99.9% across different strategies. For JjBk, they pointed to appendices for technical details, explained that Quasar leverages Python's ecosystem through external calls with optional annotations, clarified that data dependencies block parallelization naturally, and distinguished their approach from existing interpreters by emphasizing fundamentally different semantics. However, the core concern about evaluation scope remains unaddressed until promised BrowseComp-Plus full results materialize. The preliminary results on 80 of 800 tasks show similar patterns but are insufficient to resolve generalizability questions. The philosophical defense that restricted expressiveness suffices for code actions lacks empirical backing across diverse scenarios. While technical questions were answered satisfactorily, the broader concern about whether this work demonstrates sufficient practical impact beyond controlled benchmarks persists.

**Reviewer Scores:**

If reviewers had fully participated in discussion, FyqM would likely maintain their score of 6 given their low confidence and satisfaction with technical clarifications. M2ub might remain at 6 or potentially drop to 4 given the incomplete BrowseComp-Plus evaluation and unresolved scope concerns that formed their primary criticism. JjBk would likely stay at 4 since their fundamental questions about flexibility, distinction from specialized interpreters, and practical scalability received philosophical rather than empirical resolution. MYLk cannot meaningfully update from 6 given their inability to assess the work.

---

### Decision · Program_Chairs · 2026-01-26

Reject